# FASTER LANGUAGE MODELS WITH BETTER MULTI-TOKEN PREDICTION USING TENSOR DECOMPOSITION

## ABSTRACT

We propose a new model for multi-token prediction in transformers, aiming to enhance sampling efficiency without compromising accuracy. Motivated by recent work that predicts the probabilities of subsequent tokens using multiple heads, we connect this approach to rank-1 canonical tensor decomposition. By generalizing it to a rank-$r$ canonical probability decomposition, we develop an improved model that predicts multiple tokens simultaneously. This model can also be interpreted as a mixture of experts, allowing us to leverage successful techniques from that domain for efficient and robust training. Importantly, the overall overhead for training and sampling remains low. Our method demonstrates significant improvements in inference speed for both text and code generation tasks, proving particularly beneficial within the self-speculative decoding paradigm. It maintains its effectiveness across various model sizes and training epochs, highlighting its robustness and scalability.

## 1 INTRODUCTION

Autoregressive transformer models (Vaswani, 2017) have become a cornerstone in natural language processing tasks due to their ability to model complex sequential data. However, one significant limitation of these models is the inefficiency in sampling during inference, as they generate tokens one at a time, leading to increased latency in practical applications (Fournier et al., 2023; Fields et al., 2024). Accelerating the inference process without compromising the model's performance is thus a critical challenge.

Recent efforts have explored multi-token prediction to address this inefficiency. A simple yet effective approach (Gloeckle et al., 2024) involves using multiple heads to predict the next $n$ tokens simultaneously. This method approximates the joint probability of the next $n$ tokens by assuming conditional independence given the previous context. Mathematically, given a sequence $(x_1, x_2, \ldots, x_t)$ this approximation can be expressed as:

$$P_\theta(x_{t+n:t+1}|x_{t:1}) \approx \prod_{s=1}^{n} P_\theta^{(s)}(x_{t+s}|x_{t:1}). \tag{1}$$

This equation represents a rank-1 tensor approximation of the joint probability distribution, effectively treating future tokens as independent of each other given the past tokens. While this assumption simplifies computation and can be combined with speculative decoding (Leviathan et al., 2023) to accept some of the predicted tokens, it remains a crude approximation that may limit token acceptance rates due to its disregard for token interdependencies.

To improve upon this, we propose a more accurate approximation of the joint distribution by introducing a sum over multiple rank-1 terms. Specifically, we generalize the approximation to a rank-$r$ canonical decomposition (Harshman, 1970; Kolda & Bader, 2009; Cichocki et al., 2016):

$$P_\theta(x_{t+n:t+1}|x_{t:1}) \approx \sum_{\alpha=1}^{r} w_\alpha \prod_{s=1}^{n} P_\theta^{(s)}(x_{t+s}|x_{t:1}, \alpha), \tag{2}$$

where $w_\alpha \geq 0$ are learnable weights satisfying $\sum_{\alpha=1}^{r} w_\alpha = 1$.

Figure 1: Schematic representation of the proposed model that predicts several tokens at once for a given sequence $x_1, x_2, \ldots, x_t$. We present the case of $n = 3$ predicted tokens $x_{t+1}, x_{t+2}, x_{t+3}$ and, accordingly, three heads which generate factor matrices $P_\theta^{(1)}$, $P_\theta^{(2)}$, and $P_\theta^{(3)}$ of the canonical decomposition and linear layer that generates weights $w$ are depicted.

The proposed formulation in equation 2 accounts for dependencies among future tokens by effectively considering a mixture of expert models, each capturing different aspects of the token distribution. By leveraging this rank-$r$ decomposition, we aim to enhance the accuracy of multi-token predictions, thereby increasing token acceptance rates during speculative decoding and reducing overall inference time. Thus, our main contributions are as follows:

- We identify the limitations of existing multi-token prediction methods that predict tokens independently.
- We introduce a novel model that employs a rank-$r$ canonical probability decomposition to better approximate the joint distribution of future tokens.
- We demonstrate that our approach can be integrated into existing transformer architectures with minimal overhead, resulting in more efficient sampling without significant increases in computational cost.

## 2 METHOD

### 2.1 OVERALL CONCEPT

We propose a model architecture that differs from traditional transformer models by enabling simultaneous prediction of multiple tokens through a rank-$r$ Canonical Polyadic (CP) tensor decomposition (Harshman, 1970) of the joint probability distribution. In Figure 1 we provide a corresponding schematic illustration, the content of which will be disclosed later in this section.

The joint probability of the next $n$ tokens given the input sequence $x_{t:1}$ can be represented as a $n$-dimensional tensor:

$$A \in \mathbb{R}^{V \times V \times \ldots \times V}, \quad A[x_{t+1}, \ldots, x_{t+n}] = P_\theta(x_{t+n:t+1}|x_{t:1}), \tag{3}$$

where $V$ is the vocabulary size. The tensor $A$ encapsulates the probabilities of all possible combinations of the next $n$ tokens. In Gloeckle et al. (2024) it was proposed to approximate this joint distribution by assuming that future tokens are conditionally independent given the past as shown in equation 1. We draw special attention to the fact that this may be interpreted as a rank-1 CP approximation to the tensor $A$. While computationally efficient, such approximation ignores dependencies among the future tokens.

To better capture these dependencies, we propose to approximate the joint distribution using a rank-$r$ CP tensor decomposition according to equation 2. In order to ensure that $P_\theta$ from this equation is indeed a probability tensor, it is sufficient to undertake that

$$w_\alpha \geq 0, \quad \sum_{\alpha=1}^{r} w_\alpha = 1. \tag{4}$$

The difference between equation 2 and standard CP-decomposition is an additional constraint on the factors of decomposition, i.e., each factor, $P_\theta^{(s)}$ should be non-negative and sum up to 1 along one

mode:

$$\sum_{x_{t+s}=1}^{V} P_\theta^{(s)}(x_{t+s}|x_{t:1}, \alpha) = 1, \quad s = 1, 2, \ldots, n. \tag{5}$$

This is easily achieved by taking *softmax* operation along the mode direction.

Thus, for the given input sequence $x_{t:1}$ we compute its embeddings $e_{t:1}$ using the encoder of the autoregressive transformer model. Focusing on the last embedding $e_t$, we aim to predict the next $n$ tokens by parametrizing the factors of the decomposition as simple functions of $e_t$. We introduce $n$ heads each corresponding to one of the next $n$ tokens. For each position $s = 1, 2, \ldots, n$ the conditional probabilities are defined as:

$$P_\theta^{(s)}(x_{t+s}|x_{t:1}, \alpha) = \text{softmax}\left(W_\alpha^{(s)} e_t\right)_{x_{t+s}}, \tag{6}$$

where $W_\alpha^{(s)} \in \mathbb{R}^{V \times E}$ are the weight matrices for each head and component, $V$ is the vocabulary size and $E$ is the embedding dimension. The mixture weights $w_\alpha$ are computed in a similar way using an additional linear layer:

$$w = \text{softmax}\left(W_h e_t\right), \tag{7}$$

where $W_h \in \mathbb{R}^{r \times E}$.

## 2.2 Training procedure

In training, we maximize the log-likelihood of the predicted $n$ tokens. The computation of the log-likelihood is straightforward: first, the embeddings are calculated by the transformer backbone (it has the same cost as for the next token prediction). We need to evaluate the logarithm of the likelihood, so using equation 6 directly is not numerically stable. Instead, we compute everything using the logarithms of the probabilities. For each pair of sequences $x_{t:1}$ and $x_{t+n:t+1}$, we evaluate the logarithm of the mixture weights $w$ (the computational cost corresponds to a matrix-by-matrix product and logsoftmax operation), then use equation 6 to compute $n$ matrices of the size $V \times r$

$$C_{\theta,\alpha}^{(s)} = \log P_\theta^{(s)}(x_{t+s}|x_{t:1}, \alpha), \tag{8}$$

to calculate logarithms of the conditional probabilities in a stable way with *logsumexp* operation:

$$L = \log\left(P_\theta(x_{t+n:t+1}|x_{t:1})\right) \approx \log\left(\sum_{\alpha=1}^{r} w_\alpha \prod_{s=1}^{n} P_\theta^{(s)}(x_{t+s}|x_{t:1}, \alpha)\right) =$$

$$= \log\left(\sum_{\alpha=1}^{r} w_\alpha \prod_{s=1}^{n} \exp(C_{\theta,\alpha}^{(s)})\right) = \log\left(\sum_{\alpha=1}^{r} \exp\left(\log w_\alpha + \sum_{s=1}^{n} C_{\theta,\alpha}^{(s)}\right)\right). \tag{9}$$

## 2.3 Auxilary load balancing loss

Each term of the summation in equation 9 corresponds to a single **expert**, which predicts its own probabilities for each token. We have found, that optimizing such loss directly leads to the effects, similar to the ones observed in Mixture Of Experts (MoE) framework (Masoudnia & Ebrahimpour, 2014; Cai et al., 2024): one expert (i.e., rank-1 term in our case) dominates the others, leading to worser likelihood even in the presence of larger number of parameters. Note, that such interpretation and connection is not well-known in the low-rank approximation community, and can be investigated further on. To obtain the balance between different experts, we utilize the achievements from the MoE communities and propose to use an auxiliary balancing loss on $w$.

It is well known that a critical challenge in training MoE models is ensuring equitable utilization of all experts (Zhou et al., 2022). Without proper balancing, some experts may become dominant, handling a disproportionate share of the data, while others remain underutilized. To address this, we incorporate an **auxiliary balancing loss**. This auxiliary loss penalizes imbalances in the expert weights and encourages to distribute the workload evenly across all experts.

Formally, the auxiliary loss can be represented as:

$$\mathcal{L}_{\text{aux}} = \sum_{\alpha=1}^{r} \left(\frac{n_\alpha}{N} - \frac{1}{r}\right)^2, \tag{10}$$

where: $r$ is the number of experts, $n_\alpha$ is the number of tokens with maximal weight on expert $\alpha$, and $N$ is the total number of tokens. This formulation ensures that each expert $\alpha = 1, 2, \ldots, r$ is utilized approximately equally, mitigating the risk of certain experts becoming bottlenecks.

Empirical observations have demonstrated that training the model **without the auxiliary loss** or **using the auxiliary loss values proposed in previous works** leads to training instability and eventual failure. The auxiliary loss is pivotal in maintaining a balanced distribution of token assignments among experts, which is essential for stable convergence and effective learning. Therefore, careful tuning of the auxiliary loss coefficient is necessary to achieve optimal performance. By ensuring balanced expert utilization through the auxiliary loss, the model enhances the accuracy of multi-token predictions, which increases token acceptance rates during speculative decoding, thereby reducing overall inference time.

## 2.4 SAMPLING METHOD

Our sampling scheme is similar to the one proposed in Gloeckle et al. (2024). We sample candidates from the proposal distribution (our approximation to the joint distribution of the next tokens) and then accept them or reject according to the recommendations of the draft model (which is the same model that predicts the next token).

For the rank-1 case the sampling is easy: probability distributions are computed for each token independently, and sampling is done from the computed distributions. For our canonical rank-$r$ representation we need to use sequential sampling which is autoregressive, but only works with the factors of decompositions. This makes sampling $dim$ tokens from our rank $r$ model just a bit slower, than 1 token from the base model.

Note that the first marginal distribution $P(x_{t+1}|x_{t:1})$ is given by the first head directly, and we just need to average among $\alpha$:

$$P_\theta(x_{t+1}) = \sum_{\alpha=1}^{r} w_\alpha P_\theta^{(1)}(x_{t+1}|x_{t:1}, \alpha), \tag{11}$$

which can be also computed using logsumexp operation. From this distribution, we sample the first token $x_{t+1}$.

Given $x_{t+1}$ we can now compute the marginal distribution:

$$P_\theta(x_{t+2}|x_{t+1}) = \sum_{\alpha=1}^{r} w_\alpha P_\theta^{(1)}(x_{t+1}|x_{t:1}, \alpha) P_\theta^{(2)}(x_{t+2}|x_{t:1}, \alpha), \tag{12}$$

which is also reduced to matrix-by-matrix products, *logsoftmax* and *logsumexp* operations, and can be implemented by updating the unnormalized logits of the experts with incorporation of $\log P_\theta^{(1)}(x_{t+1}|x_{t:1}, \alpha)$ into them.

The sampling of the following tokens is also straightforward. Given sampled $x_{t+1}, \ldots, x_{t+s-1}$ we then compute the probability:

$$P_\theta(x_{t+s}|x_{t+1}, \ldots x_{t+s-1}) = \sum_{\alpha=1}^{r} w_\alpha \prod_{k=1}^{s-1} P_\theta^{(k)}(x_{t+k}|x_{t:1}, \alpha). \tag{13}$$

## 2.5 SPECULATIVE DECODING

Speculative decoding (Chen et al., 2023; Leviathan et al., 2023) is a technique designed to accelerate the inference process of autoregressive models by generating multiple tokens in parallel, thereby reducing the latency associated with sequential token generation. In traditional autoregressive sampling, tokens are generated one at a time, with each new token conditioning on the previously generated tokens. This sequential nature inherently limits the speed of generation, especially for lengthy outputs.

Our sampling method seamlessly integrates with the speculative decoding framework by enhancing its capacity to handle multi-token predictions, as can be seen from Algorithm 1. The usual setup

---

**Algorithm 1** Self-speculative decoding with rank-r experts

---

**Require:** prefix $X$, encoder $E$, weight function $W$, heads $H_i$, dim $n$, rank $r$
1: $e_t \leftarrow E(X)[-1]$ {Take last embedding}
2: $w_t \leftarrow log(\text{softmax}(W(e_t)))$ {Obtain expert weights}
3: $LP_t \leftarrow [H_i(e_t) : 0 \leq i \leq r]$ {Obtain log core probabilities by expert}
4: $S_t, P_t \leftarrow RankRSample(w_t, LP_t)$ {Sample from rank $r$ head as described in equation 2. $S_t$ are the samples and $P_t$ are the conditional probability distributions}
5: $e_p^{\text{list}} \leftarrow E(X)[-n :]$ {Obtain new embeddings}
6: accept $\leftarrow True$
7: $i \leftarrow 0$
   {We use the scheme as in Leviathan et al. (2023) below}
8: **while** accept **do**
9: $\quad u \leftarrow Uniform(0, 1)$
10: $\quad P_p^i \leftarrow FirstHeadPrediction(e_p^i, W, H_1)$ {Obtain probability distribution from first head as described in equation 2}
11: $\quad c \leftarrow P_p^i[S_t^{i+1}]/P_t^{i+1}[S_t^{i+1}]$ {Different indexes due to offset induced by the fact, that first token by our draft model is always from the same distribution, as it would be from a base model}
12: $\quad$ **if** $u < c$ **then**
13: $\quad\quad i \leftarrow i + 1$
14: $\quad$ **else**
15: $\quad\quad$ accept $\leftarrow False$
16: $\quad$ **end if**
17: **end while**
18: **if** $i < n$ **then**
19: $\quad P_{last} \leftarrow \text{normalize}(max(0, P_p^i - P_t^{i+1}))$
20: **else**
21: $\quad P_{last} \leftarrow FirstHeadPrediction(e_p^n, W, H_1)$
22: **end if**
23: $s_l \leftarrow \text{sample}(P_{last})$ {additional sample from base model}
24:
25: **return** prefix $+ S_t[: i + 1] + s_l$

---

for a speculative decoding consists of a draft model and a base model. In our case we implement a modification of a self-speculative decoding algorithm, as described in Zhang et al. (2023); Elhoushi et al. (2024). So, as a base model we take a next-token prediction model and as a draft model – the prediction for $dim$ tokens forward obtained from a full "CP-head".

Self speculative decoding with rank $r$ model inherits two nice small benefits from more simple rank-1 model: first generated sample from the draft model is always accepted and one additional token from the base model is generated. This means, that in one pass of the draft model with the base model we will obtain at least 2 tokens. Due to this fact it seems beneficial to use that type of models even with moderate quality of the draft model.

## 3 EXPERIMENTS

In this section, we present a comprehensive evaluation of our proposed multi-token prediction approach. Experiments are designed to assess the efficacy of different ranks and auxiliary loss configurations, the capability to fine-tune only the prediction head, and the impact on inference speed for large-scale models.

### 3.1 TRAINING DIFFERENT RANKS AND AUXILIARY LOSS MODELS

For our experiments we've chosen the multi-head tiny transformer model with 56.3 M parameters based on the code in Karpathy (2022). We consider the case of 4 heads and added RoPe positional encodings as in Su et al. (2024). Training was conducted on the Tiny Stories dataset (Eldan & Li,

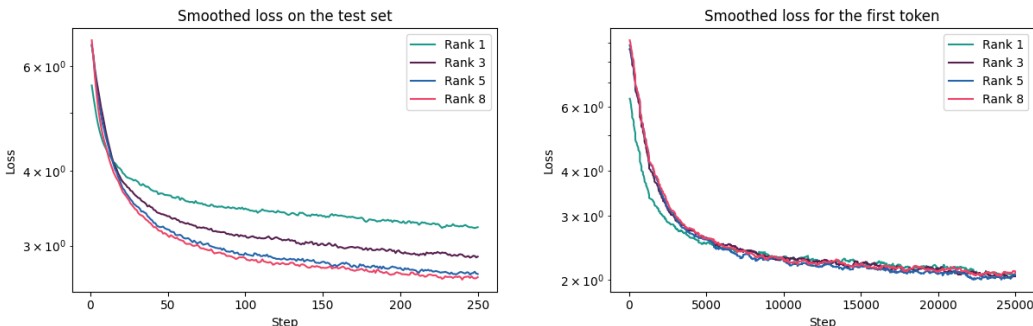

Figure 2: Losses for the tiny transformer model with different CP-rank values trained on the TinyStories dataset.

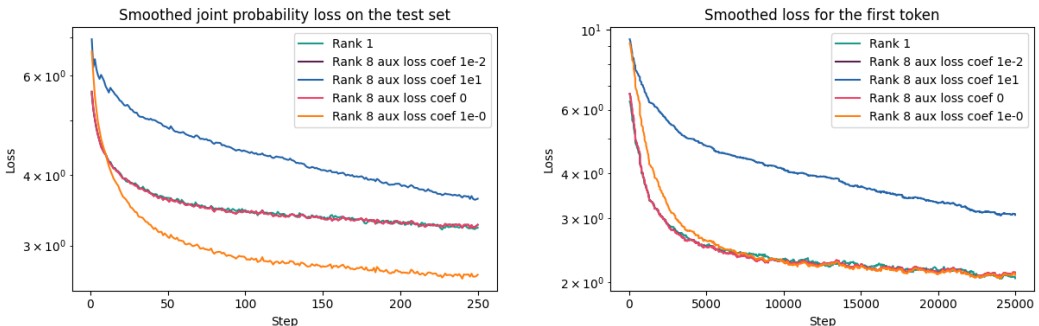

Figure 3: Losses for the rank-8 tiny transformer model trained on the TinyStories dataset with different auxiliary loss penalties compared to the baseline (i.e., the rank-1 model).

2023) using various ranks for the CP-decomposition. The objective was to observe how increasing the rank influences the joint loss and loss on the first token. Because the quality of our final generation depends only on the quality of the first head, we tracked both those metrics. Additionally, we experimented with different sizes of the auxiliary loss penalty to ensure balanced expert utilization.

As illustrated in the left graph in Figure 2, increasing the rank from 1 to higher values leads to a consistent decrease in joint loss, indicating a better approximation of the joint probability distribution. This trend underscores the model's enhanced capability to capture inter-token dependencies with higher ranks.

Contrary to the joint loss, right graph in Figure 3 shows that the loss for the first token remains largely unchanged across different ranks. It is worth noting, that probability distribution for the first token as a function of last layer embeddings $m$ is given by $\sum_{\alpha=1}^{r} w_\alpha(m) C_\alpha(m)$ (in notation of equation 9) and both $C_\alpha$ and $w_\alpha$ are linear, which makes this function equivalent to a simple linear head. So, after convergence we expected the same loss for all of the ranks. As follows from the reported results, this is exactly what happened and this consistency confirms that our model maintains optimal training for the initial token prediction, ensuring that the foundational aspects of the sequence generation remain robust. The loss on the first token is especially crucial, because with a speculative decoding we are improving sample for a big model, which is in our self-speculative case is a first head. We also note that from Figure 2 it follows that all of the inference speedup will be obtained without compromising quality.

Figure 3 presents the effect of varying the auxiliary loss penalty size. We observed that with a very small penalty, the joint loss mirrors that of the rank-1 model, suggesting insufficient balancing among experts. Conversely, an excessively large penalty led to prolonged convergence times, as depicted in the figure. Then we identified an optimal penalty size balancing expert utilization without hampering training accuracy.

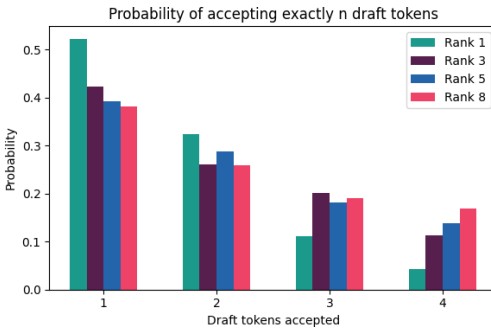 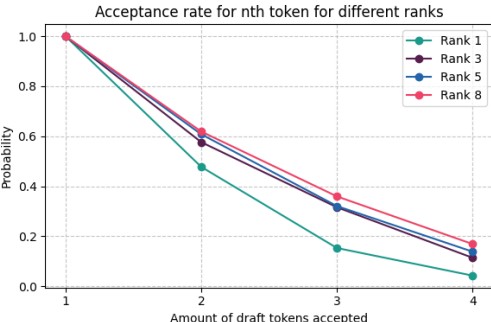

Figure 4: Speculative decoding performance for the tiny transformer model with different CP-rank values trained on the TinyStories dataset from scratch.

Table 1: Results with the speculative decoding for the tiny transformer model with different CP-rank values trained on the TinyStories dataset from scratch.

| Rank | Loss | Avg. draft tokens accepted | Time per token (with speculative decoding) |
|------|------|----------------------------|--------------------------------------------|
| 1 | 3.23 | 1.67 | 0.0336s |
| 3 | 2.88 | 2.01 | 0.0328s |
| 5 | 2.69 | 2.07 | **0.0303s** |
| 8 | 2.66 | 2.15 | 0.0326s |

Table 2: Average number of accepted draft tokens for the PyCode model.

| Rank | Loss | Average Draft Tokens Accepted |
|------|------|-------------------------------|
| 1 | 2.07 | 1.52 |
| 3 | 1.88 | 1.64 |
| 5 | 1.80 | 1.65 |

The efficiency of our model in speculative decoding was evaluated by measuring the acceptance rate of drafted tokens. Figure 4 and Table 1 illustrates that on the Tiny Stories dataset, models with higher ranks achieved up to a around 30% increase in accepted drafts. This allowed us to reduce inference time even for this tiny ("nanoGPT") model for which the head is responsible for a significant percentage of computational time, which is not the case for larger models.

## 3.2 HEAD-ONLY FINE-TUNING FOR PYCODE MODEL

To evaluate the flexibility of our approach, we fine-tuned only the prediction head of the Py-CodeGPT (Zan et al., 2022) model across different ranks on the Github Code dataset by CodePar-rot. [1] This experiment aimed to determine whether partial model updates could yield performance improvements without the computational overhead of full model fine-tuning.

Figure 5 and Table 2 demonstrates speculative decoding performance for the experiments we con-ducted for different rank values. From the reported results it follows that even when only the head is fine-tuned, increasing the rank leads to marginal improvements in joint loss. Additionally, we can see that speculative decoding benefits from higher ranks, albeit to a lesser extent (approximately 9% increase in accepted drafts) compared to the full model training.

## 3.3 INFERENCE TIME BENCHMARKING

To determine the impact of modified head on the inference time of bigger language models we benchmarked the time of one forward pass of our approach on large-scale models with 3 billion and 8 billion parameters. As reported in Table 3, the inference overhead for integrating the proposed

---

[1]See https://huggingface.co/datasets/codeparrot/github-code.

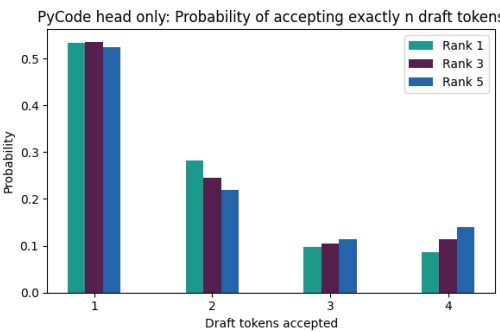 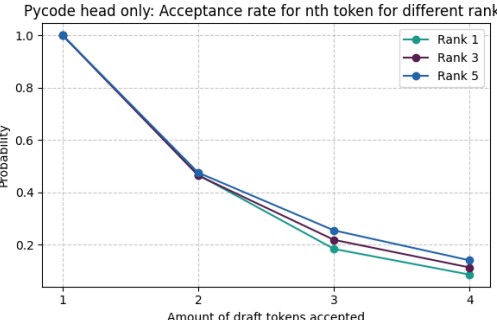

Figure 5: Speculative decoding performance for trained head of the PyCode transformer model with different CP-rank values.

Table 3: Inference time for one forward pass comparison for Llama and Rocket models.

| Rank | Llama 8B Barebone | Llama 8B Head | Llama 8B Full | Rocket 3B Full |
|---|---|---|---|---|
| Barebone | 0.1761 | - | 0.1761s | 0.0154 |
| Rank 1 | 0.1761 | 0.0132 | 0.1893 | 0.0160 |
| Rank 3 | 0.1825 | 0.0129 | 0.1954 | 0.0162 |
| Rank 5 | 0.1865 | 0.0330 | 0.2195 | 0.0166 |

multi-head layer remains minimal, even as the rank increases. Note that for the value of CP-rank of 5 we observe a significant time increase for head execution then the Llama model is considerd which is probably caused by its huge vocabulary size. However, for moderate-sized networks inference time remains limited and increases only slightly with increasing CP-rank. The obtained results correspond to the theoretical algorithmic complexity of our new layer. During inference computational complexity of barebone grows linearly (given KV caches), but computational complexity of rank-$r$ head is always the same. Our measurements were made with seq length varying from 1024 to 4096, but for many practical applications sequence length is bigger, which further justifies usage of rank-$r$ head in the case of models with a large context window.

## 4 RELATED WORK

Training large language models (LLMs) to predict multiple tokens all at once and in parallel can drive these models toward better sample efficiency. Various approaches for multi-token predictions have been proposed recently. In Stern et al. (2018) several feed-forward decoder layers from the last encoder state are added for prediction of the next several tokens simultaneously, and in Miao et al. (2024) this idea was further improved within the framework of the so-called Medusa heads that use tree attention mechanism. In a number of works Song et al. (2021); Santilli et al. (2023); Fu et al. (2024) it is proposed to generate multiple draft tokens in parallel on the basis of the Jacobi iteration methods, i.e., via solving a non-linear system of equations while auto-regressive decoding in LLM. In the work Bhendawade et al. (2024) multiple tokens are predicted by adding streaming embeddings initialized from upper layers, with the token tree reduced by early exiting.

Thus, this direction of research is actively developing today, however, the approaches outlined above have several limitations, including the need for significant changes in the original architecture of the model and limited speedup. Therefore, of particular interest is the recent work Gloeckle et al. (2024), where it was proposed to approximate the joint probability of the next several tokens using multiple heads but assuming conditional independence given the previous context. As we have already noted above, this approach remains a crude approximation that may limit token acceptance rates in the speculative decoding approach due to its disregard for token interdependencies. To improve upon this, we considered in this work a more accurate approximation of the joint distribution in the form of the CP-decomposition.

To effectively implement the proposed scheme, we paid attention to connection of the used weighted CP-decomposition with the Mixture of Experts (MoE) technique. MoE is a widespread approach to enhance capabilities of LLMs with the most popular one being Sparse-Gated MoE introduced in Shazeer et al. (2017). MoE implementations can be either sparse or dense with sparse version being more popular, but there are many usages of both options, as in Dou et al. (2023) and Pan et al. (2024). While many parameters of MoE approach can be tweaked (Cai et al., 2024), the most common option is using MoE inside a transformer block, as in Zhou et al. (2022). We also note that MoE usage is not limited to LLMs and, for example, in Oldfield et al. (2024) it is applied to computer vision model.

In this work, as an application of the proposed model for multi-token prediction, we consider its use as part of the speculative decoding scheme, which was proposed in Leviathan et al. (2023) and nowadays has become a common technique in the domain of inference acceleration. While initial framework solves the problem of inference optimization of a model given a faster draft model, there are different methods to obtain this draft model. Early works proposed blockwise decoding as in Stern et al. (2018). This line of work is similar to ours, as the model, used for speculative decoding, is exactly the same, as base model. Later more techniques for self-speculative decoding were developed, namely in Elhoushi et al. (2024) it is proposed to use only particular layers of the base model to obtain draft model and in Hooper et al. (2023) the base model consists of cycles, which also allows to skip layers to obtain a draft model. Self speculative decoding and multi-token prediction naturally go well with each other. This combination may require modification in model architecture as in Bhendawade et al. (2024), but it is possible to modify only heads as in Gloeckle et al. (2024) to enable faster application of the approach to existing LLMs, and we use such approach in our work.

## 5  CONCLUSION

In this work, we propose a new model for multi-token prediction in transformers based on the Canonical Polyadic (CP) tensor decomposition of the joint probability distribution. The results indicate that our model can be efficiently trained across a wide range of ranks, with higher ranks consistently yielding lower joint losses. This improvement underscores the model's ability to better capture the dependencies among future tokens, leading to more accurate predictions.

We observed a direct correlation between lower joint losses and enhanced speculative decoding performance. Specifically, our approach significantly increased the acceptance rates of predicted tokens, with notable improvements of up to 50 % of draft tokens accepted. The factor matrices of our decomposition of the joint probability tensor are generated by several heads that use shared model trunk, which practically makes it possible to minimize extra costs during inference and convert higher draft token acceptance to faster inference times.

The ability to fine-tune only the prediction head of the model while maintaining competitive performance highlights the flexibility of our approach. This capability allows for targeted improvements without the computational overhead associated with full model retraining. Benchmarking inference speed for bigger models demonstrated that our method introduces negligible inference overhead, ensuring that in many practical cases the benefits of improved performance for draft model do not come at the cost of increased latency.

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
