# OpenReview forum: "Faster Language Models with Better Multi-Token Prediction Using Tensor Decomposition"
_ICLR.cc/2025/Conference — Submitted to ICLR 2025_

### Official Review · Reviewer_Juyo · 2024-10-31

**Soundness:** 2
**Presentation:** 3
**Contribution:** 3
**Rating:** 5
**Confidence:** 4

**Summary:**

This paper focuses on speculative decoding methods that incorporate additional prediction heads. The authors conceptualize current standard approaches as rank-1 canonical tensor decomposition and propose a generalized method that extends from rank-1 to rank-r canonical tensor decomposition to approximate the joint distribution of future tokens. To enhance model training, an auxiliary loss is introduced to address weight imbalances. Experimental results highlight several key findings:

1.	Increasing the ranks results in a decrease in joint loss.

2.	The first token appears to have no correlation with different ranks.

3.	The method is effective even when only the prediction heads are trained.

The proposed approach achieves notable speedups compared to autoregressive models and rank-1 baselines.

**Strengths:**

1.	This work identifies the limitations of recent multi-token prediction standards and proposes a more generalized approach.

2.	The experimental results demonstrate the method's effectiveness, and the ablation study underscores the importance of the introduced components.

**Weaknesses:**

1.	The work lacks comparison with existing state-of-the-art methods such as Medusa, Eagle, etc., which belong to the same research domain.

2.	In the code generation setting, the performance of averaging two accepted draft tokens is not promising.

3.	There are several typos in this version that need revision.

**Questions:**

1.	In line 113, the authors denote the input sequence as x_{t:1} and the corresponding embeddings as e_{t:1}. According to the description, the embeddings are the representations of the final transformer layer, while in Figure 1, the same value is denoted as z_t. Do z_t and e_t means the same representation, or e_t means the “input” embeddings? This notation is somewhat confusing.

2.	Are there any results on the acceptance rate for Llama 8B, not just inference time?

__Typos__:

1.	In line 116, a comma is missing before "the conditional probabilities ...".
2.	In line 150, "Note, that" should be revised to "Note that".

---

> ### Author Response · Authors · 2024-11-23
>
> 1. The work lacks comparison with existing state-of-the-art methods such as Medusa, Eagle, etc., which belong to the same research domain.
>
>
> We've considered our method as a generalization of the method, introduced in the paper [1], so we've taken this method as a baseline and made a comparison only with a base model (no speculative decoding) and rank 1 model (model from paper [1])
>
> 2. In the code generation setting, the performance of averaging two accepted draft tokens is not promising.
>
> We agree that the performance achieved on the PyCode experiment is not particularly strong. However, we would like to emphasize that in this case, we trained only the heads, leaving the main model unchanged. The current results indicate that for the cost of two forward passes, we generate approximately 2.5 tokens, representing a speedup over the vanilla model. While there is room for improvement, we believe this demonstrates the potential of our approach, even in a limited setting.
>
>
> 3. In line 113, the authors denote the input sequence as $x_{t:1}$ and the corresponding embeddings as $e_{t:1}$. According to the description, the embeddings are the representations of the final transformer layer, while in Figure 1, the same value is denoted as $z_t$. Do $z_t$ and $e_t$ means the same representation, or $e_t$ means the “input” embeddings? This notation is somewhat confusing.
>
> You are correct, and we appreciate your attention to this detail. $z_t$ and $e_t$ refer to the same object in this case. To avoid confusion, we have updated Figure 1 to use $e_t$ consistently throughout the manuscript.
>
> 4. Are there any results on the acceptance rate for Llama 8B, not just inference time?
>
> The inference times for Llama are obtained for untrained heads; we have not trained the model for this task as it is computationally expensive. Inference time is presented to estimate the performance of one forward pass of a draft model with our custom head.
>
>
>
> We've also conducted an additional set of experiment, that involved training of a 1B model on a fineweb dataset. To obtain the table, which can be seen below, we've generated 2048 tokens 10 times and averaged the results between different runs.
>
> | Generation of 2048 tokens | Time (s) | Amnt of parameters (B) | Average number of accepted tokens |
> |------------------------|-----------|---------------|------------|
> | base                   | 43.8      | -             | -          |
> | rank 1                 | 34        | 0.98          | 1.58       |
> | rank 2                 | 28.9      | 1.2           | 1.96       |
> | rank 4                 | 31.5      | 1.9           | 1.85       |

---

### Official Review · Reviewer_5c1w · 2024-11-04

**Soundness:** 2
**Presentation:** 2
**Contribution:** 2
**Rating:** 5
**Confidence:** 4

**Summary:**

One existing form of speculative decoding involves predicting k tokens at a time independently, which can be thought of as a rank-1 decomposition of the k-order joint probability tensor over those tokens. This paper instead proposes to predict the factors for a rank-r decomposition. They evaluate two instantiations of this idea: training a LM from scratch to predict this decomposition, and taking an existing LM and fine-tuning additional heads to predict this decomposition. Their experiments show that higher rank decompositions lead to higher acceptance rates in speculative decoding.

**Strengths:**

(1) The method is well-motivated and explained clearly. The connection to MoE, which motivates a load-balancing auxiliary loss, is also interesting.

(2) The paper seeks to improve inference speed in large models, which is an important problem.

**Weaknesses:**

While the method seems interesting and promising, the paper's experiments seem disorganized and insufficient to fully demonstrate the effectiveness of the method.

(1) The majority of the results are for a 56.3M parameter trained on TinyStories, which is a very limited evaluation setting, both because the dataset is synthetic and because the setting involves retraining. There are also some experiments on head-only tuning for PyCodeGPT in Table 3, but the results in that setting are not very strong --- increasing the rank does not actually seem to actually improve inference speed for many of the models. The paper would benefit from more thorough evaluation and stronger results (especially on non-synthetic datasets, and on speeding up existing models rather than requiring retraining: for example, the evaluations done in https://arxiv.org/pdf/2211.17192 (Table 3) would improve this paper).

(2) The majority of the experiments section seems to involve analysis rather than results: only tables 1 and 3 report inference times, which are the main results. I would suggest moving other plots (token acceptance rate, first token vs joint loss, etc.) to a separate analysis section.

(3) There are a substantial number of issues with the experiment design that would be beneficial to address: (a) In Figure 3, it seems like hyperparameters are being selected using the test set; I would suggest using a dev set instead. (b) To make comparisons fair, I would suggest training each rank for the same amount of wall-clock time, rather than number of steps, in case higher ranks require more time per forward pass. (c) The self-speculative setup makes the results hard to interpret because each rank uses a different target model. I would suggest that each method be speculative with respect to the same target model. (d) The paper would be clearer if the experiments were described concretely: for example, the paper states that "Our measurements were made with seq length varying from 1024
to 4096" (lines 408-409), but it's not clear which experiments use which sequence lengths.

**Questions:**

See above.

---

> ### Author Response · Authors · 2024-11-23
>
> 1. The paper would benefit from more thorough evaluation and stronger results.
>
> Thank you for the detailed and constructive feedback! In Table 3 in the provided paper we see the calculation of $\alpha$ for different pairs of target and draft models. However, we think, that in our framework calculating $\alpha$ makes no practical sense for two reasons. The first one is the fact, that probabilities of accepting prefixes of different lengths are i.i.d (paragraph under definition 3.1 in the original paper). This assumption makes perfect sense in the case of usual speculative decoding, but this is not true in our case. Secondly, in practice, $\alpha$ allows to calculate of an optimal amount of generated draft tokens. This knowledge is basically useless in our case because once we trained the model with some $d$, it makes no sense to sample less, than $d$ tokens from it.
>
> To address concerns about broader evaluations, we have conducted additional experiments with a 1B parameter model trained on the fineweb dataset.  For inference we've generated 2048 tokens 10 times and averaged the results between runs.  These results are provided in the updated table below, showcasing our approach's scalability and applicability to non-synthetic datasets.
>
> | Generation of 2048 tokens | Time (s) | Amnt of parameters (B) | Average number of accepted tokens |
> |------------------------|-----------|---------------|------------|
> | base                   | 43.8      | -             | -          |
> | rank 1                 | 34        | 0.98          | 1.58       |
> | rank 2                 | 28.9      | 1.2           | 1.96       |
> | rank 4                 | 31.5      | 1.9           | 1.85       |
>
>
> 2. The majority of the experiments section seems to involve analysis rather than results.
>
> Thank you for that suggestion. We considered such a division but decided to leave the presentation of the results as they are. In our case, dividing the results into parts, as you suggested, would have led to unnecessary fragmentation, which in turn would have made the article more difficult to understand.
>
> 3. In Figure 3, it seems like hyperparameters are being selected using the test set; I would suggest using a dev set instead.
>
> You're right, it's an oversight on our part. However, we tune this parameter only on the one dataset, and for other tasks, we've used the same penalty on auxiliary loss. Also, the results do not change, when we use the separate dev set.
>
> 4. To make comparisons fair, I would suggest training each rank for the same amount of wall-clock time, rather than a number of steps, in case higher ranks require more time per forward pass.
>
> In small-scale experiments, we trained until convergence, so adding additional iterations to the base model won't change the result. In any case, we haven't claimed any speedups to the training process itself.
>
>
> 5. The self-speculative setup makes the results hard to interpret because each rank uses a different target model. I would suggest that each method be speculative with respect to the same target model.
>
> You're right, technically the statement 'our model accepts more tokens per forward pass than a baseline model' is meaningless. However, specifically for that reason, we provide a chart, which can be seen on the right of Figure 2. Here one can see, that our trained models achieve basically the same loss when we're looking at the first token only. This means, that the target model for each case has the same loss, which, in our opinion makes the comparison fair.
>
>
> 6. The paper would be clearer if the experiments were described concretely: for example, the paper states that "Our measurements were made with seq length varying from 1024 to 4096" (lines 408-409), but it's not clear which experiments use which sequence lengths.
>
> Thank you for highlighting this ambiguity. For the specific example mentioned, we sampled 1,000 random sequence lengths from the interval [1024, 4096] and ran the model on prefixes of these lengths. We will revise the manuscript to describe the experimental setup in more concrete terms for all relevant cases.
>
> We hope these responses address your concerns and clarify the points you raised. Your suggestions have been instrumental in improving the clarity and rigor of our work, and we sincerely appreciate your thorough review. Please let us know if there are additional aspects you would like us to address.

---

> > ### Comment · Reviewer_5c1w · 2024-12-02
> >
> > Thanks for your detailed response! The new results on fineweb look promising, and more results of that form would make the paper stronger. I have increased my score to a 5, but I still think that the paper needs better organization and more complete evaluation.

---

### Official Review · Reviewer_8dbJ · 2024-11-04

**Soundness:** 3
**Presentation:** 3
**Contribution:** 2
**Rating:** 5
**Confidence:** 3

**Summary:**

After reading the author response, I thank the authors for providing clarifications to Table 1 and Table 3 and answering some of my other questions. However, I still think it is important to compare to other baselines (e.g. EAGLE, Medusa as another reviewer points out) in addition to Gloeckle et al. 2024. Therefore I am keeping my score the same.

----

The paper studies multi token prediction in transformer language models. Vanilla autoregressive degressing is expensive for long outputs since it only decodes one output at a time.

The authors are inspired by the work of Gloeckle et al. 2024. In Gloeckle et al. 2024, given a context x_{t:1} the next n tokens are predicted independently (with multiple heads). As the authors point out, this amounts to a rank-1 tensor approximation of the joint probability distribution.

In this work, the authors explore higher ranks (r > 1) using CP decomposition. They draw a connection to mixture-of-experts and propose a auxiliary load balancing strategy so all the weight is not on one expert (component).

They then perform experiments validating their work.

**Strengths:**

-Tackles an interesting and important problem

-The method is written clearly.

-I also find the connections to tensor decomposition interesting.

**Weaknesses:**

Some confusion on experimental results: I'm a bit confused as to how much of a speed up the author's approach gives over both the approach of Goeckle et al. 2024 (i.e. rank=1) and also vanilla non-autoregressive decoding for the same level of quality.

For example in Table 1: I see that in Table 1 the  final column (time per token) is not much different across all the rows?

Moreover I don't quite understand Table 3.

Comparisons: I think the authors need additional baselines in addition to just ablations of their own approach from the related work. For example, as another reviewer suggested EAGLE and Medusa:
https://github.com/SafeAILab/EAGLE
https://arxiv.org/abs/2401.10774

Related work: The authors should also cite and discuss related work in non-autoregressive decoding (typically for neural machine translation) that has been developed for a while e.g. see below and citations therein. In particular it would be useful to discuss how the authors' approach compares and contrasts with these works.

https://arxiv.org/abs/1711.02281
https://arxiv.org/abs/2204.09269
https://arxiv.org/abs/2012.15833

**Questions:**

-How does the method combine with beam search?

-Does the speedup increase or decrease as a function of model size?

---

> ### Author Response · Authors · 2024-11-23
>
> Dear reviewer, thank you very much for your analysis of our work and the specific notes you formulated! Below we provide our responses to them. We will be happy to answer any additional questions you may have.
>
> 1 ... how much of a speed up the author's approach gives over both the approach of Goeckle et al. 2024 (i.e. rank=1) and also vanilla non-autoregressive decoding for the same level of quality.
>
> Thank you for raising this important point. Table 1 in the original submission includes a comparison with the baseline approach. However, we recognize that it lacked a direct comparison with vanilla non-autoregressive decoding. To address this, we have conducted additional experiments using the fineweb dataset, which has this essential comparison. We've trained 1B base model from scratch using different ranks. For inference we've generated 2048 tokens 10 times and averaged the results between runs. The results can be seen in the table below.
>
> | Generation of 2048 tokens | Time (s) | Amnt of parameters (B) | Average number of accepted tokens |
> |------------------------|-----------|---------------|------------|
> | base                   | 43.8      | -             | -          |
> | rank 1                 | 34        | 0.98          | 1.58       |
> | rank 2                 | 28.9      | 1.2           | 1.96       |
> | rank 4                 | 31.5      | 1.9           | 1.85       |
>
>
>
> 2. ...  in Table 1 the final column (time per token) is not much different across all the rows?
>
> In this column, we observe a 10 percent speedup when compared to a baseline speculative decoding approach. When compared to the vanilla model, the acceleration is much greater, as you can see in the table above.
>
> 3.  I don't quite understand Table 3.
>
> In this table, we are showing, how replacing a vanilla head with our proposed head affects the performance of one forward pass of the model. As we can see, as the model size increases, the overhead becomes less noticeable.
>
> 4.  ... the authors need additional baselines ...
>
> As our approach is a generalization of the approach, introduced in work [1], we compared our approach only with this baseline.
>
>
> 5. ... cite and discuss related work in non-autoregressive decoding
>
>
> Thank you for adding the necessary context for our work. We were mainly focused on the task of text generation, so we overlooked those works. We'll cite those works, as they are relevant to our topic.
>
>
> 6. How does the method combine with beam search?
>
> Our method inherits the same limitations as the approach introduced in [1]. It is fully compatible with commonly used text generation strategies, including sampling with temperature and top-k candidate selection, which we have implemented in our experiments. While we have not yet tested beam search specifically, we see no theoretical obstacles to combining our method with it. We plan to explore this combination in future work.
>
> 7. Does the speedup increase or decrease as a function of model size?
>
> Typically, as the model size increases, the ratio (computational complexity of head)/(computational complexity of the entire model) becomes smaller. This benefits our method, as our modification allows us to make fewer forward passes with the cost of a more computationally expensive head.

---

> > ### Comment · Reviewer_8dbJ · 2024-11-25
> >
> > Thank you for the response and the clarifications to Table 1 and 3. However, I still think comparisons to other baselines are needed besides just Goeckle et al. 2024 (i.e. rank=1) e.g. EAGLE/Medusa as another other reviewer points out.
> >
> > Moreover, I think the improved speedup over the rank-1 model is also modest overall.

---

### Official Review · Reviewer_MLew · 2024-11-09

**Soundness:** 3
**Presentation:** 3
**Contribution:** 3
**Rating:** 5
**Confidence:** 4

**Summary:**

This paper borrows key idea from Gloecke et al. [1] to train multi-token predictors instead of single next word predictor. This work identifies a key flaw in [1] which is that the distributions for multiple $n$ future tokens are independent of each other thus ignoring the token  interdependency. This work interprets this as a rank-1 approximation to the full distribution tensor of $n$ next tokens and proposes to improve this to a higher rank estimate. This higher rank estimate is achieved by $r$ multiple heads defining $r$ different distributions and using their mixture for the $n$-future token prediction. Training and inference method for this is discussed followed by an observation that the multi-token predictor can be used in self-speculative sampling approach where the next word prediction is made faster by using proposal distribution that predicts multiple next tokens. The experiments are mainly performed on nano-GPT model architecture trained on TinyStories dataset and also finetuning the PyCodeGPT model.

**Strengths:**

-- The paper studies an interesting problem to speed updecoding by predicting multiple tokens in parallel at higher acceptance rates than typical speculative sampling approaches.

-- The proposed solution seems straightforward to implement.

-- The contribution to identifying issues with existing multi-token training approaches and proposing a higher rank alternative is novel.

**Weaknesses:**

-- The evaluation leaves a lot to be desired. Experiments are done on small datasets and small models but more concerningly, little else is provided aside from loss curves of training runs and token acceptance rates for the scheduled sampling approach. As an example, performance of these models on various benchmarks to estimate the quality of these trained models would aid in better assessment of the approach. Also, it is unclear if this approach empirically scaled to larger datasets and models effectively in terms of speed and performance.

-- Comparison to other speculative sampling approaches with various draft models will give abetter idea about the improvement on speed and resources with the proposed approach.

-- There is room for improvement in presentation. Figure 1 doesn't help with understanding the paper better and is confusing. Algorithm 1 can also be described more clearly. Currently, it hinges on the reader's prior understanding of speculative decoding.

**Questions:**

Please address the issues above.

---

> ### Author Response · Authors · 2024-11-23
>
> Dear reviewer, Thank you very much for analyzing our work and formulating the specific questions you asked! Below are our responses to your questions.
>
> 1.  Experiments are done on small datasets and small models...
>
> To address your concerns, we've conducted an additional set of experiments on a 1B model using fineweb as a dataset. The results are in the attached table. We've generated 2048 tokens 10 times and averaged the results between different runs.
>
> | Generation of 2048 tokens | Time (s) | Amnt of parameters (B) | Average number of accepted tokens |
> |------------------------|-----------|---------------|------------|
> | base                   | 43.8      | -             | -          |
> | rank 1                 | 34        | 0.98          | 1.58       |
> | rank 2                 | 28.9      | 1.2           | 1.96       |
> | rank 4                 | 31.5      | 1.9           | 1.85       |
>
>
>
> 2. ...  little else is provided aside from loss curves of training runs and token acceptance rates for the scheduled sampling approach...
>
> We've also provided time per token for the tinystories dataset. Experiments with fineweb also contain time per token. To the best of our knowledge, these are the most relevant metrics for evaluating our approach. If there are other specific metrics you'd like us to consider, we would be happy to explore them.
>
> 3. ... performance of these models on various benchmarks to estimate the quality of these trained models ...
>
> We've trained different models on different datasets to ensure the compatibility of our method with different tasks. Overall, our method is a generalization of the method from [1], so we compare the performance of our novel method with this previous one.
>
> 4. Comparison to other speculative sampling approaches with various draft models will give a better idea ...
>
> At the current scale, alternative approaches seem faster. However, as the model scale increases, the approach [1] becomes more and more effective. And as our approach has better performance, than then approach from [1], we believe, that our approach also has a good performance scaling with model size.
>
> 5. There is room for improvement in presentation ...
>
>
> Thank you for this feedback! Our goal is to make our paper as accessible, as it is possible. Algorithm 1 mainly describes the compatibility of our new proposed draft model with the existing approach, therefore, the description is not as detailed as in the sources in which the speculative decoding approach was first described. As for Figure 1, we believe, that it answers the question "How n-dimensional probability distribution over $R^n$ is parameterized by the set of $2-$dimensional matrices". We think, that both Algorithm 1 and Figure 1 are essential to the narrative we're constructing in this article. However, we can add additional figures, especially if there are specific issues that can be explained graphically.
>
>
> We hope our responses address your concerns and improve the clarity and completeness of our manuscript. Please let us know if there are any other aspects we should focus on. Thank you again for your valuable feedback!

---

> > ### Comment · Reviewer_MLew · 2024-11-26
> > **Thanks for your rebuttal**
> >
> > Hello, thanks for responding with some additional experiments on 1B models. My overall impression about the paper and the empirical comparison remains the same so I am keeping my score unchanged.

---

### Meta-Review · Area_Chair_93L7 · 2024-12-22

**Metareview:**

This paper aims to speed up the generation process of language models. Motivated by the observation that predicting multiple future tokens in parallel does not consider inter-token dependencies, and that it can be viewed as rank-1 tensor decomposition, this work proposes to extend that to a rank-r decomposition. Experiments show that the proposed method has higher acceptance rates compared to the baseline.

Strengths:
1. The proposed idea is well motivated and the extension to rank-r decomposition is neat.

Weaknesses:
1. Reviewers mentioned that experiments are only conducted on small models with small datasets, but this is addressed by authors' rebuttal.
2. Reviewers mentioned that baselines are lacking and this work should be compared to EAGLE and Medusa.
3. Some reviewers mentioned that the presentation of this paper can be improved.

Overall, the idea is very interesting and the proposed extension to rank-r decomposition is very neat. However, experiments can be further improved by adding more contemporary baselines, and the new larger experiments authors conducted during rebuttal can be added to the paper. I'm recommending rejection based on reviewers' opinions, but I wouldn't mind if the paper gets accepted.

**Additional Comments On Reviewer Discussion:**

Two common concerns are: 1. the experiments conducted seem not sufficient in scale and analyses. This has been addressed by authors during rebuttal. 2. Baselines are lacking (such as EAGLE and Medusa), and this seems not addressed yet.

---

### Decision · Program_Chairs · 2025-01-22

Reject